# Multi-View Silhouette and Depth Decomposition for High Resolution 3D Object Representation

**Edward Smith**
McGill University
edward.smith@mail.mcgill.ca

**Scott Fujimoto**
McGill University
scott.fujimoto@mail.mcgill.ca

**David Meger**
McGill University
dmeger@cim.mcgill.ca

## Abstract

We consider the problem of scaling deep generative shape models to high-resolution. Drawing motivation from the canonical view representation of objects, we introduce a novel method for the fast up-sampling of 3D objects in voxel space through networks that perform super-resolution on the six orthographic depth projections. This allows us to generate high-resolution objects with more efficient scaling than methods which work directly in 3D. We decompose the problem of 2D depth super-resolution into silhouette and depth prediction to capture both structure and fine detail. This allows our method to generate sharp edges more easily than an individual network. We evaluate our work on multiple experiments concerning high-resolution 3D objects, and show our system is capable of accurately predicting novel objects at resolutions as large as $512 \times 512 \times 512$ – the highest resolution reported for this task. We achieve state-of-the-art performance on 3D object reconstruction from RGB images on the ShapeNet dataset, and further demonstrate the first effective 3D super-resolution method.

## 1 Introduction

The 3D shape of an object is a combination of countless physical elements that range in scale from gross structure and topology to minute textures endowed by the material of each surface. Intelligent systems require representations capable of modeling this complex shape efficiently, in order to perceive and interact with the physical world in detail (e.g., object grasping, 3D perception, motion prediction and path planning). Deep generative models have recently achieved strong performance in hallucinating diverse 3D object shapes, capturing their overall structure and rough texture [3, 37, 47]. The first generation of these models utilized voxel representations which scale cubically with resolution, limiting training to only $64^3$ shapes on typical hardware. Numerous recent papers have begun to propose high resolution 3D shape representations with better scaling, such as those based on meshes, point clouds or octrees but these often require more difficult training procedures and customized network architectures.

Our 3D shape model is motivated by a foundational concept in 3D perception: that of canonical views. The shape of a 3D object can be completely captured by a set of 2D images from multiple viewpoints (see [21, 4] for an analysis of selecting the location and number of viewpoints). Deep learning approaches for 2D image recognition and generation [40, 10, 8, 13] scale easily to high resolutions. This motivates the primary question in this paper: *can a multi-view representation be used efficiently with modern deep learning methods?*

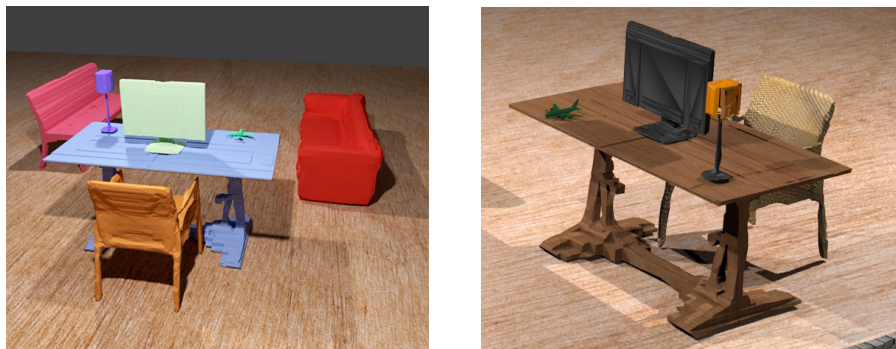

Figure 1: Scene created from objects reconstructed by our method from RGB images at $256^3$ resolution. See the supplementary video for better viewing `https://sites.google.com/site/mvdnips2018`.

We propose a novel approach for deep shape interpretation which captures the structure of an object via modeling of its canonical views in 2D as depth maps, in a framework we refer to as Multi-View Decomposition (MVD). By utilizing many 2D orthographic projections to capture shape, a model represented in this fashion can be up-scaled to high resolution by performing semantic super-resolution in *2D space*, which leverages efficient, well-studied network structures and training procedures. The higher resolution depth maps are finally merged into a detailed 3D object using model carving.

Our method has several key components that allow effective and efficient training. We leverage two synergistic deep networks that decompose the task of representing an object's depth: one that outputs the silhouette – capturing the gross structure; and a second that produces the local variations in depth – capturing the fine detail. This decomposition addresses the blurred images that often occur when minimizing reconstruction error by allowing the silhouette prediction to form sharp edges. Our method utilizes the low-resolution input shape as a rough template which simply needs carving and refinement to form the high resolution product. Learning the residual errors between this template and the desired high resolution shape simplifies the generation task and allows for constrained output scaling, which leads to significant performance improvements.

We evaluate our method's ability to perform 3D object reconstruction on the the ShapeNet dataset [1]. This standard evaluation task requires generating high resolution 3D objects from single 2D RGB images. Furthermore, due to the nature of our pipeline we present the first results for 3D object super-resolution – generating high resolution 3D objects directly from low resolution 3D objects. Our method achieves state-of-the-art quantitative performance, when compared to a variety of other 3D representations such as octrees, mesh-models and point clouds. Furthermore, our system is the first to produce 3D objects at $512^3$ resolution. We demonstrate these objects are visually impressive in isolation, and when compared to the ground truth objects. We additionally demonstrate that objects reconstructed from images can be placed in scenes to create realistic environments, as shown in figure 1. In order to ensure reproducible experimental comparison, code for our system has been made publicly available on a GitHub repository[1]. Given the efficiency of our method, each experiment was run on a single NVIDIA Titan X GPU in the order of hours.

## 2 Related Work

**Deep Learning with 3D Data**    Recent advances with 3D data have leveraged deep learning, beginning with architectures such as 3D convolutions for object classification [25, 19]. When adapted to 3D generation, these methods typically use an autoencoder network, with a decoder composed of 3D deconvolutional layers [3, 47]. This decoder receives a latent representation of the 3D shape and produces a probability for occupancy at each discrete position in 3D voxel space. This approach has been combined with generative adversarial approaches [8] to generate novel 3D objects [47, 41, 20], but only at a limited resolution.

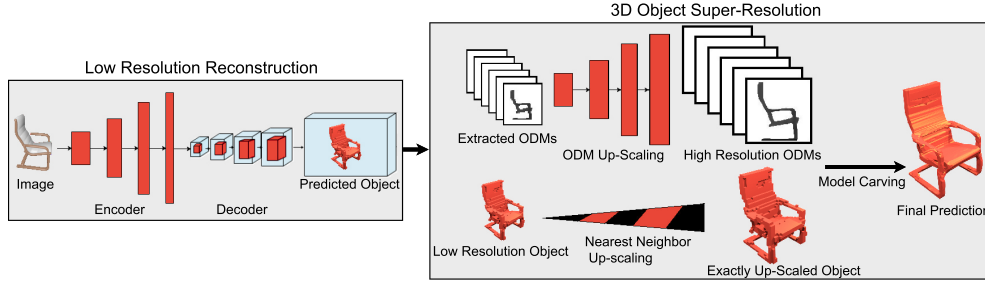

Figure 2: The complete pipeline for 3D object reconstruction and super-resolution outlined in this paper. Our method accepts either a single RGB image for low resolution reconstruction or a low resolution object for 3D super-resolution. ODM up-scaling is defined in section 3.1 and model carving in section 3.2

**2D Super-Resolution** Super-resolution of 2D images is a well-studied problem [29]. Traditionally, image super-resolution has used dictionary-style methods [7, 49], matching patches of images to higher-resolution counterparts. This research also extends to depth map super-resolution [22, 28, 11]. Modern approaches to super-resolution are built on deep convolutional networks [5, 46, 27] as well as generative adversarial networks [18, 13] which use an adversarial loss to imagine high-resolution details in RGB images.

**Multi-View Representation** Our work connects to multi-view representations which capture the characteristics of a 3D object from multiple viewpoints in 2D [17, 26, 43, 32, 12, 39, 34], such as decomposing image silhouettes [23, 42], Light Field Descriptors [2], and 2D panoramic mapping [38]. Other representations aim to use orientation [36], rotational invariance [15] or 3D-SURF features [16]. While many of these representations are effective for 3D classification, they have not previously been utilized to recover 3D shape in high resolution.

**Efficient 3D Representations** Given that naïve representations of 3D data require cubic computational costs with respect to resolution, many alternate representations have been proposed. Octree methods [44, 9] use non-uniform discretization of the voxel space to efficiently capture 3D objects by adapting the discretization level locally based on shape. Hierarchical surface prediction (HSP) [9] is an octree-style method which divides the voxel space into free, occupied and boundary space. The object is generated at different scales of resolution, where occupied space is generated at a very coarse resolution and the boundary space is generated at a very fine resolution. Octree generating networks (OGN) [44] use a convolutional network that operates directly on octrees, rather than in voxel space. These methods have only shown novel generation results up to $256^3$ resolution. Our method achieves higher accuracy at this resolution and can efficiently produce novel objects as large as $512^3$.

A recent trend is the use of unstructured representations such as mesh models [31, 14, 45] and point clouds [33, 6] which represent the data by an unordered set with a fixed number of points. MarrNet [48], which resembles our work, models 3D objects through the use of 2.5D sketches, which capture depth maps from a single viewpoint. This approach requires working in voxel space when translating 2.5D sketches to high resolution, while our method can work directly in 2D space, leveraging 2D super-resolution technology within the 3D pipeline.

## 3 Method

In this section we describe our methodology for representing high resolution 3D objects. Our algorithm is a novel approach which uses the six axis-aligned orthographic depth maps (ODM), to efficiently scale 3D objects to high resolution without directly interacting with the voxels. To achieve this, a pair of networks is used for each view, decomposing the super-resolution task into predicting the silhouette and relative depth from the low resolution ODM. This approach is able to recover fine object details and scales better to higher resolutions than previous methods, due to the simplified learning problem faced by each network, and scalable computations that occur primarily in 2D image space.

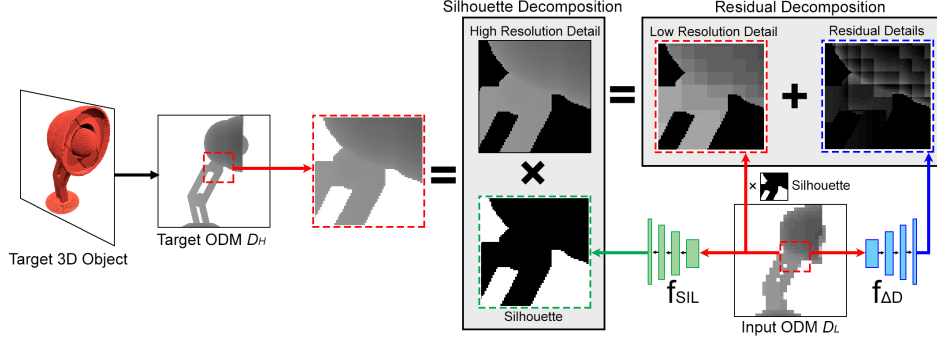

Figure 3: Our Multi-View Decomposition framework (MVD). Each ODM prediction task can be decomposed into a silhouette and detail prediction. We further simplify the detail prediction task by encoding only the residual details (change from the low resolution input), masked by the ground truth silhouette.

## 3.1 Orthographic Depth Map Super-Resolution

Our method begins by obtaining the orthographic depth maps of the six primary views of the low-resolution 3D object. In an ODM, each pixel holds a value equal to the surface depth of the object along the viewing direction at the corresponding coordinate. This projection can be computed quickly and easily from an axis-aligned 3D array via z-clipping. Super-resolution is then performed directly on these ODMs, before being mapped onto the low resolution object to produce a high resolution object.

Representing an object by a set of depth maps however, introduces a challenging learning problem, which requires both local and global consistency in depth. Furthermore, minimizing the mean squared error results in blurry images without sharp edges [24, 30]. This is particularly problematic as a depth map is required to be bimodal, with large variations in depth to create structure and small variations in depth to create texture and fine detail. To address this concern, we propose decomposing the learning problem into predicting the silhouette and depth map separately. Separating the challenge of predicting gross shape from fine detail regularizes and reduces the complexity of the learning problem, leading to improved results when compared with directly estimating new surface depths.

Our Multi-View Decomposition framework (MVD) uses a set of twin of deep convolutional models $f_{\text{SIL}}$ and $f_{\Delta\text{D}}$, to separately predict silhouette and variations in depth of the higher resolution ODM. We depict our system in figure 3. The deep convolutional network for predicting the high-resolution silhouette, $f_{\text{SIL}}$ with parameters $\theta$, is passed the low resolution ODM $D_L$, extracted from the input 3D object. The network outputs a probability that each pixel is occupied. It is trained by minimizing the mean squared error between the predicted and true silhouette of the high resolution ODM $D_H$:

$$\mathcal{L}(\theta) = \sum_{i=1}^{N} \|f_{\text{SIL}}(D_L^{(i)}; \theta) - \mathbb{1}_{D_H^{(i)} \neq 0}(D_H^{(i)})\|_2, \tag{1}$$

where $\mathbb{1}_{D_H^{(i)} \neq 0}$ is an indicator function for each coordinate in the image.

The same low-resolution ODM $D_L$ is passed through the second deep convolution neural network, denoted $f_{\Delta\text{D}}$ with parameters $\phi$, whose final output is passed through a sigmoid, to produce an estimate for the variation of the ODM within a fixed range $r$. This output is added to the low-resolution depth map to produce our prediction for a constrained high-resolution depth map $C_H$:

$$C_H = r\sigma(f_{\Delta\text{D}}(D_L; \phi)) + g(D_L), \tag{2}$$

where $g(\cdot)$ denotes up-sampling using nearest neighbor interpolation.

We train our network $f_{\Delta\text{D}}$ by minimizing the mean squared error between our prediction and the ground truth high-resolution depth map $D_H$. During training only, we mask the output with the ground truth silhouette to allow effective focus on fine detail for $f_{\Delta\text{D}}$. We further add a smoothing regularizer which penalizes the total variation $V(x) = \sum_{i,j} \sqrt{(x_{i+1,j} - x_{i,j})^2 + (x_{i,j+1} - x_{i,j})^2}$ [35] within

the predicted ODM. Our loss function is a simple combination of these terms:

$$\mathcal{L}(\phi) = \sum_{i=1}^{N} \|(C_H^{(i)} \circ \mathbb{1}_{D_H^{(i)}(j,k)\neq 0}(D_H^{(i)})) - D_H^{(i)}\|_2 + \lambda V(C_H^{(i)}), \qquad (3)$$

where $\circ$ is the Hadamard product. The total variation penalty is used as an edge-preserving denoising which smooths out irregularities in the output.

The output of the constrained depth map and silhouette networks are then combined to produce a complete prediction for the high-resolution ODM. This accomplished by masking the constrained high-resolution depth map by the predicted silhouette:

$$\hat{D}_H = C_H \circ f_{\text{SIL}}(D_L;\theta). \qquad (4)$$

$\hat{D}_H$ denotes our predicted high resolution ODM which can then be mapped back onto the original low resolution object by model carving to produce a high resolution object. Each of the 6 high resolution ODMS are predicted using the same 2 network models, with the side information for each passed using a forth channel in the corresponding low resolution ODM passed to the networks.

### 3.2  3D Model Carving

To complete our super-resolution procedure, the six ODMs are combined with the low-resolution object to form a high-resolution object. This begins by further smoothing the up-sampled ODM with an adaptive averaging filter, which only consider neighboring pixels within a small radius. To preserve edges, only neighboring pixels within a threshold of the value of the center pixel are included. This smoothing, along with the total variation regularization in the our loss function, are added to enforce smooth changes in local depth regions.

Model carving begins by first up-sampling the low-resolution model to the desired resolution, using nearest neighbor interpolation. We then use the predicted ODMs $\hat{D}_H = C_H \circ f_{\text{SIL}}(D_L;\theta)$ to determine the surface of the new object. The carving procedure is separated into (1) structure carving, corresponding to the silhouette prediction $f_{\text{SIL}}(D_L;\theta)$, and (2) detail carving, corresponding to the constrained depth prediction $C_H$.

For the structure carving, for each predicted ODM $f_{\text{SIL}}(D_L;\theta)$, if a coordinate is predicted unoccupied, then all voxels perpendicular to the coordinate are highlighted to be removed. The removal only occurs if there is agreement of at least two ODMs for the removal of a voxel. As there is a large amount of overlap in the surface area that the six ODMs observe, this silhouette agreement is enforced to maintain the structure of the object.

This same process occurs for detail carving with $C_H$. However, we do not require agreement within the constrained depth map predictions. This is because, unlike the silhouettes, a depth map can cause or deepen concavities in the surface of the object which may not be visible from any other face. Requiring agreement among depth maps would eliminate their ability to influence these concavities. Thus, performing detail carving simply involves removing all voxels perpendicular to each coordinate of each ODM, up to the predicted depth.

## 4  Experiments

In this section we present our results for our method, Multi-View Decomposition Networks (MVD), for both 3D object super-resolution and 3D object reconstruction from single RGB images. Our results are evaluated across 13 classes of the ShapeNet [1] dataset. 3D super-resolution is the task of generating a high resolution 3D object conditioned on a low resolution input, while 3D object reconstruction is the task of re-creating high resolution 3D objects from a single RGB image of the object.

### 4.1  3D Object Super-Resolution

**Dataset**  The dataset consists of $32^3$ low resolution voxelized objects and their $256^3$ high resolution counterparts. These objects were produced by converting CAD models found in the ShapeNetCore dataset [1] into voxel format, in a canonical view. We work with the three commonly used object

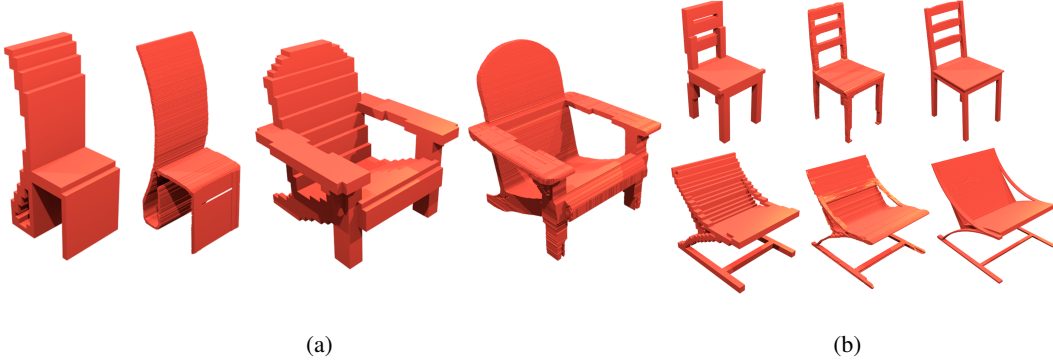

(a)                                                     (b)

Figure 4: Super-resolution rendering results. Each set shows, from left to right, the low resolution input and the results of MVD at $512^3$. Sets in (b) additionally show the ground-truth $512^3$ objects on the far right.

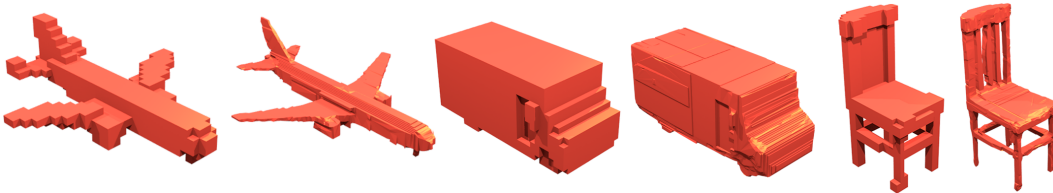

Figure 5: Super-resolution rendering results. Each pair shows the low resolution input (left) and the results of MVD at $256^3$ resolution (right).

classes from this dataset: Car, Chair and Plane, with around 8000, 7000, 4000 objects respectively. For training, we pre-process this dataset, to extract the six ODMs from each object at high and low-resolution. CAD models converted at this resolution do not remain watertight in many cases, making it difficult to fill the inner volume of the object. We describe an efficient method for obtaining high resolution voxelized objects in the supplementary material. Data is split into training, validation, and test set using a ratio of 70:10:20 respectively.

**Evaluation**   We evaluate our method quantitatively using the intersection over union metric (IoU) against a simple baseline and the prediction of the individual networks on the test set. The baseline method corresponds to the ground truth at $32^3$ resolution, by up-scaling to the high resolution using nearest neighbor up-sampling. While our full method, MVD, uses a combination of networks, we present an ablation study to evaluate the contribution of each separate network.

**Implementation**   The super-resolution task requires a pair of networks, $f_{\triangle D}$ and $f_{SIL}$, which share the same architecture. This architecture is derived from the generator of SRGAN [18], a state of the art 2D super-resolution network. Exact network architectures and training regime are provided in the supplementary material.

**Results**   The super-resolution IoU scores are presented in table 1. Our method greatly outperforms the naïve nearest neighbor up-sampling baseline in every class. While we find that the silhouette prediction contributes far more to the IoU score, the addition of the depth variation network further increases the IoU score. This is due to the silhouette capturing the gross structure of the object from multiple viewpoints, while the depth variation captures the fine-grained details, which contributes less to the total IoU score. To qualitatively demonstrate the results of our super-resolution system we render objects from the test set at both $256^3$ resolution in figure 5 and $512^3$ resolution in figure 4. The predicted high-resolution objects are all of high quality and accurately mimic the shapes of the ground truth objects. Additional $512^3$ renderings as well as multiple objects from each class at $256^3$ resolution can be found in our supplementary material.

## 4.2   3D Object Reconstruction from RGB Images

**Dataset**   To match the datasets used by prior work, two datasets are used for 3D object reconstruction, both derived from the ShapeNet dataset. The first, which we refer to as $Data_{HSP}$, consists of

| Category | Baseline | Depth Variation ($f_{\Delta D}$) | Silhouette ($f_{SIL}$) | MVD (Both) |
|----------|----------|----------------------------------|-------------------------|------------|
| Car | 73.2 | 80.6 | 86.9 | **89.9** |
| Chair | 54.9 | 58.5 | 67.3 | **68.5** |
| Plane | 39.9 | 50.5 | 70.2 | **71.1** |

Table 1: Super-Resolution IoU Results against nearest neighbor baseline and an ablation over individual networks at $256^3$ from $32^3$ input.

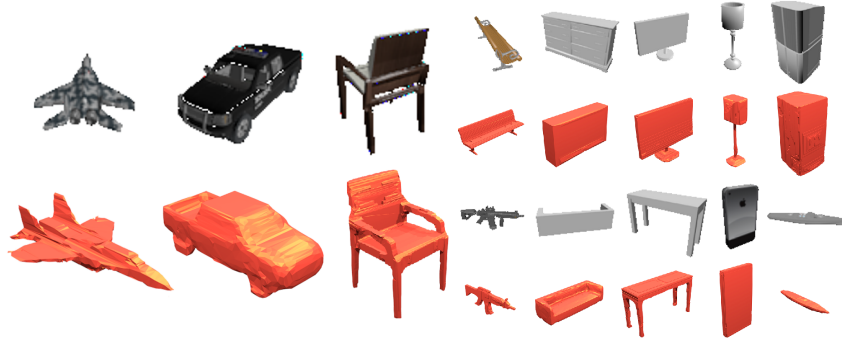

Figure 6: 3D object reconstruction $256^3$ rendering results from our method, MVD (bottom), of the 13 classes from ShapeNet, by interpreting 2D image input (top).

only the Car, Chair and Plane classes from the Shapenet dataset, and we re-use the $32^3$ and $256^3$ voxel objects produced for these classes in the previous section. The CAD models for each of these object were rendered into $128^2$ RGB images capturing random viewpoints of the objects at elevations between $(-20°, 30°)$ and all possible azimuth rotations. The voxelized objects and corresponding images were split into a training, validation and test set, with a ratio of 70:10:20 respectively.

The second dataset, which we refer to as $Data_{3D-R2N2}$, is that provided by Choy et al. [3]. It consists of images and objects produced from the 3 classes in the ShapeNet dataset used in the previous section, as well as 10 additional classes, for a total of around 50000 objects. From each object $137^2$ RGB images are rendered at random viewpoints, and we again compute their $32^3$ and $256^3$ resolution voxelized models and ODMs. The data is split into a training, validation and test set with a ratio of 70:10:20.

**Evaluation** We evaluate our method quantitatively with two evaluation schemes. In the first, we use IoU scores when reconstructing objects at $256^3$ resolution. We compare against HSP [9] using the first dataset $Data_{HSP}$, and against OGN [44] using the second dataset $Data_{3D-R2N2}$. To study the effectiveness of our super-resolution pipeline, we also compute the IoU scores using the low resolution objects predicted by our autoencoder (AE) with nearest neighbor up-sampling to produce predictions at $256^3$ resolution.

Our second evaluation is performed only on the second dataset, $Data_{3D-R2N2}$, by comparing the accuracy of the surfaces of predicted objects to those of the ground truth meshes. Following the evaluation procedure defined by Wang et al. [45], we first convert the $256^3$ voxel models into meshes by defining squared polygons on all exposed faces on the surface of the voxel models. We then uniformly sample points from the two mesh surfaces and compute F1 scores. Precision and recall are calculated using the percentage of points found with a nearest neighbor in the ground truth sampling set less than a squared distance threshold of 0.0001. We compare to state of the art mesh model methods, N3MR [14] and Pixel2Mesh [45], a point cloud method, PSG [6], and a voxel baseline, 3D-R2N2 [3], using the values reported by Wang et al. [45].

**Implementation** For 3D object reconstruction, we first trained a standard autoencoder, similar to prior work [3, 41], to produce objects at $32^3$ resolution. These low resolution objects are then used with our 3D super-resolution method, to generate 3D object reconstructions at a high $256^3$ resolution. This process is described in figure 2. The exact network architecture and training regime are provided in the supplementary material.

| Category | AE | HSP [9] | MVD (Ours) | Category | AE | OGN [44] | MVD (Ours) |
|---|---|---|---|---|---|---|---|
| Car | 55.2 | 70.1 | **72.7** | Car | 68.1 | 78.2 | **80.7** |
| Chair | 36.4 | 37.8 | **40.1** | Chair | 37.6 | - | **43.3** |
| Plane | 28.9 | 56.1 | **56.4** | Plane | 34.6 | - | **58.9** |

(a) $Data_{HSP}$ (b) $Data_{3D-R2N2}$

Table 2: 3D Object Reconstruction IoU at $256^3$. Cells with a dash - indicate that the corresponding result was not reported by the original author.

| Category | 3D-R2N2 [3] | PSG [6] | N3MR [14] | Pixel2Mesh [45] | MVD (Ours) |
|---|---|---|---|---|---|
| Plane | 41.46 | 68.20 | 62.10 | 71.12 | **87.34** |
| Bench | 34.09 | 49.29 | 35.84 | 57.57 | **69.92** |
| Cabinet | 49.88 | 39.93 | 21.04 | 60.39 | **65.87** |
| Car | 37.80 | 50.70 | 36.66 | **67.86** | 67.69 |
| Chair | 40.22 | 41.60 | 30.25 | 54.38 | **62.57** |
| Monitor | 34.38 | 40.53 | 28.77 | 51.39 | **57.48** |
| Lamp | 32.35 | 41.40 | 27.97 | **48.15** | 48.37 |
| Speaker | 45.30 | 32.61 | 19.46 | 48.84 | **53.88** |
| Firearm | 28.34 | 69.96 | 52.22 | 73.20 | **78.12** |
| Couch | 40.01 | 36.59 | 25.04 | 51.90 | **53.66** |
| Table | 43.79 | 53.44 | 28.40 | 66.30 | **68.06** |
| Cellphone | 42.31 | 55.95 | 27.96 | 70.24 | **86.00** |
| Watercraft | 37.10 | 51.28 | 43.71 | 55.12 | **64.07** |
| Mean | 39.01 | 48.58 | 33.80 | 59.72 | **66.39** |

Table 3: 3D object reconstruction surface sampling F1 scores.

**Results** The results of our IoU evaluation compared to the octree methods [44, 9] can be seen in table 2. We achieve state-of-the-art performance on every object class in both datasets. Our surface accuracy results can be seen in table 3 compared to [45, 6, 14, 3]. Our method achieves state of the art results on all 13 classes. We show significant improvements for many object classes and demonstrate a large improvement on the mean over all classes when compared against the methods presented. To qualitatively evaluate our performance, we rendered our reconstructions for each class, which can be seen in figure 6. Additional renderings can be found in the supplementary material.

## 5 Conclusion

In this paper we argue for the application of multi-view representations when predicting the structure of objects at high resolution. We outline our Multi-View Decomposition framework, a novel system for learning to represent 3D objects and demonstrate its affinity for capturing category-specific shape details at a high resolution by operating over the six orthographic projections of the object.

In the task of super-resolution, our method outperforms baseline methods by a large margin, and we show its ability to produce objects as large as $512^3$, with a 16 times increase in size from the input objects. The results produced are visually impressive, even when compared against the ground-truth. When applied to the reconstruction of high-resolution 3D objects from single RGB images, we outperform several state of the art methods with a variety of representation types, across two evaluation metrics.

All of our visualizations demonstrate the effectiveness of our method at capturing fine-grained detail, which is not present in the low resolution input but must be captured in our network's weights during learning. Furthermore, given that the deep aspect of our method works entirely in 2D space, our method scales naturally to high resolutions. This paper demonstrates that multi-view representations along with 2D super-resolution through decomposed networks is indeed capable of modeling complex shapes.

## Footnotes

[1]https://github.com/EdwardSmith1884/Multi-View-Silhouette-and-Depth-Decomposition-for-High-Resolution-3D-Object-Representation

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
