[Supplementary Material]

# Supplementary Material

## A  Super-Resolution Network Architecture

Both $f_{\Delta D}$ and $f_{\text{SIL}}$ share the same architecture, which is derived from the generator of SRGAN [6], a state of the art 2D super-resolution network.

The architecture begins with a single convolutional layer followed by 16 identical residual blocks [3] with batch normalization, ReLU activations, and skip connections attaching each subsequent layer. This is followed by a convolutional layer with batch normalization, a ReLU activation function, and a skip connection attached to the output of the first convolutional layer. The final layers are a series of sub-pixel convolution layers to increase the resolution of the images [7], followed a final convolutional layer to decrease the color channel to 1, with a sigmoid activation to limit the output range. The number of sub-pixel convolution layers is equal to $\log_2$ of the upscaling factor. The kernel size is $3 \times 3$ and stride length is 1 for all layers, and all convolutional layers have kernel depth 128 except for the last, with depth 1. The kernel depth begins at size 256 for the sub-pixel convolutional layers and decreases by a factor of 2 for each subsequent layer, to offset the increase in kernel height and width.

Each network is trained using Adam [5] with default hyper-parameters and a learning rate of $10^{-3}$, trained over mini-batches of size 32, until convergence. We use a $5 \times 5$ adaptive averaging filter with a threshold of 10 for $256^3$ objects and 20 for $512^3$ objects. The output of $f_{\Delta D}$ is constrained to a maximum of $r = 70$ for $256^3$ objects and $r = 90$ for $512^3$ objects.

## B  Low Resolution Object Reconstruction Network Architecture

The network used for predicting the low-resolution 3D objects is a deep convolutional autoencoder. The encoder network of this system takes as input a RGB image and passes it through five convolutional layers with batch normalization, leaky-ReLU activations, and stride length 2, followed by a fully connected layer to produce a vector of length 128. The network architecture for the decoder begins with a fully connected layer to increase the vector to length 1024 followed by an alternation of nine 3D deconvolutional and convolutional layers to morph the up-sampled vector to a complete 3D shape. It outputs a $32^3$ matrix of voxel probabilities. Training was performed using the Adam optimizer [5], using mean squared error loss, and was halted when IoU scores on the validation set stopped decreasing.

## C  Dataset Details

A main problem with voxelizing mesh models at high resolution is that meshes may not be water tight. This is makes producing solid objects, without any unintended holes or unfilled areas, difficult. A method to fix this problem, suggested by Häne et al. [2], involves eroding one voxel across the entire surface of the lower resolution model, applying a nearest neighbor up-sampling to high-resolution, occupying all voxels that intersect with the mesh, and then applying a graph-cut based regularization with a small smoothness term to decide the remaining voxels. While this does rectify the issue of non-watertight meshes, it may not reproduce the original surface perfectly and may lead to an overly smooth model.

We suggest a new method to produce accurate, high-resolution voxel models from non-watertight CAD models. We first convert the CAD model to voxels at resolution $256^3$, and determine their orthographic depth maps. The high-resolution models are then down-sampled to $32^3$ resolution (wherein they are guaranteed to be watertight), then all internal voxels are filled, next they are up-sampled to the original resolution using nearest neighbor interpolation. Finally, the six depth faces are used to carve away the surface voxels of the reproduced high-resolution object. The only situation in which this does not make a complete model is in the rare case when the CAD model is missing one or more large faces at some point on its surface, and these objects are automatically discarded as no true voxel object can be extracted from the model, although this occurrence is rare, and does not occur in almost all object classes.

## D Analysis of Super Resolution for ODMs

Several state of the art super-resolution techniques were tested alongside our own architecture. The first was a slight variant on SRGAN [6], a state of the art adversarial generation system for image super-resolution, adept at producing photo-realistic RGB images at up to a 4 times resolution increase. The SRGAN system applies the generic GAN loss formulation [1] along side a VGG loss (based on the difference of layer activations from a pre-trained VGG network [8]) to upscale images, equipped with two deep convolutional neural networks acting as the generator and discriminator. The VGG loss term was removed from the generator loss function, and replaced by MSE loss as our dataset is far more constrained.

The second super-resolution algorithm compared was the SRGAN algorithm without adversarial loss. This corresponds to the generator of SRGAN directly predicting the higher resolution image, trained with a MSE loss. This was used as the adversarial loss is employed to achieve photorealism rather than reconstruction accuracy.

The third super-resolution scheme tested for our task was MS-Net [4], the state of the art for depth map super-resolution. This passes depth maps though a CNN consisting of a convolutional layer followed by, 3 deconvolution networks to increase the image dimensionality, then culminating in a final convolutional layer to output the high-resolution image. The novelty in the scheme is that instead of passing the image directly, only the high frequency details are passed through the network, and the result is the added to the original images low frequency information which is up-sampled to the higher resolution using bi-cubic interpolation.

We compare the accuracy of these algorithms to our own by testing their performance at recovering $256^2$ ODMs from $32^2$ ODMs from the chair object class. We also test the performance of our algorithm when omitting smoothing, not including our information from the occupancy maps, and when not including information from the depth maps. We train, validate, and test on the same 70:10:20 split as for the image reconstruction task. We trained all networks using the Adam optimizer [5] with a learning rate of $10^{-4}$, and halted learning when the performance on the validation set tested every epoch, bottomed out. The MSE for each algorithm on our held-out test set is shown in table 1 As can be seen, our algorithm achieves far lower error when recovering ODMs. The results demonstrate that smoothing and depth map information all play a role in improving the accuracy of our algorithm.

| Method | MSE |
|---|---|
| SRGAN [6] | 1268.53 |
| SRGAN Generator [6] | 919.64 |
| MS-Net [4] | 1659.89 |
| Ours (Silhouette only) | 813.28 |
| Ours (without smoothing) | 745.72 |
| Ours | **712.25** |

Table 1: Comparison of super-resolution methods via mean squared error (MSE).

# E    Super-Resolution Visualizations

Super resolution renderings for the 13 classes of ShapeNet are presented on the following pages. Images are presented in the following order: low resolution (left), super-resolution output (center) and ground truth (right).

(a) $512^3$ Chair

(b) $512^3$ Chair

(c) $256^3$ Car

(d) $256^3$ Plane

(a) $256^3$ Sofa

(b) $256^3$ Speaker

(c) $256^3$ Cellphone

(d) $256^3$ Table

(a) $256^3$ Cabinet

(b) $256^3$ Firearm

(c) $256^3$ Monitor

(d) $256^3$ Lamp

(a) $256^3$ Watercraft

(b) $256^3$ Chair

(c) $256^3$ Bench

# F    Object Reconstruction Visualizations

3D object reconstruction renderings for the 13 classes of ShapeNet are presented on the following pages. The $32^2$ image inputs are presented on the left, and the high resolution output is presented on the right.

(a) $256^3$ Plane

(b) $256^3$ Table

(c) $256^3$ Car

(d) $256^3$ Chair

(e) $256^3$ Bench

(f) $256^3$ Sofa

(g) $256^3$ Speaker

(h) $256^3$ Cellphone

(i) $256^3$ Watercraft

(j) $256^3$ Cabinet

(k) $256^3$ Firearm

(l) $256^3$ Monitor

(m) $256^3$ Lamp

# G    Failure Cases from Low Resolution Reconstruction

The following examples demonstrate how the ODM predictions react to failure in the low resolution prediction. Missing details are ignored, and poor accuracy is not exacerbated but also not remedied.

Figure 1: Examples of failure cases in object reconstruction.

# H    Ablation Study over number of ODMS

We studied the effect of reducing the number of sides to up-sample on the chair class. When an ODM was removed it was assumed that its opposite face was symmetrical and so was used in its place. As can be seen, even for a fairly symmetrical class like chairs the use of all 6 sides is important.

| Number of ODMs | 3 | 4 | 5 | 6 |
|---|---|---|---|---|
| IoU Score | 61.8 | 62.9 | 65.2 | **68.5** |

Table 2: Ablation study over number of ODMs on IoU Accuracy on the Chair Class at $256^3$ Resolution.