[Reviews · NeurIPS 2018]

Reviewer 1



Summary: This paper proposes a 3d super resolution method. It first projects the reconstructed 3D model to 6 low resolution orthographic depth maps. A mask network and a depth network are trained to up-sample the corresponding depth maps to high resolution ones, and a 3D model carving strategy is applied to produce high resolution reconstructed result. In their experiments, the reconstructed results are out perform the previous state-of-the-art pix2mesh algorithm by first do a low resolution reconstruction & then do super-resolution. Novelty: The insight of the paper is leverage the difficulty of super-resolution in 3D to the field of well studied image super-resolution, so that the learning could be much easier. Several questions seems not presented in the paper: 1: Is it still necessary to estimate the view point of the object in order to align the model to canonical views for up-sampling networks ? 2: For single image reconstruction task, the super-resolution in 3D is independent with image after low-res reconstruction. how is the result when the single image low resolution reconstruction fails, will the super-resolution enlarge the error ? Maybe some failure case show be shown. 3: In L87, "[41] show the results up to 256^3". I think this is because the dataset has ground truth at the resolution of 256^3, while the method has the capability to extend to 512^3 (as also shown in their paper), and they have the code released. The author should show a comparison on the same resolution (either visually or numerically). Another problem of the paper is that the description is not good, e.g. L116: structure -> structural L125: passed -> passed in ? the math symbols to represent different contents are somewhat sloppy ( what is the j, k in Eqn. (1) and (2), why depth map for high resolution is \hat{D}_H while for low resolution is C_L? rather than \hat{D}_L). Experiments: The ablation study of how many canonical view (ODM) is needed w.r.t the final performance ? Since most object are symmetric in shapenet, maybe 3 is already good enough ? How is the visual comparison between the proposed method with [41] ? It is important to have the intuition of the place of improvement for supporting the numbers. Overall, although the method gives impressive results on reconstruction, I still have concern about the representation could be limited in real-world images since canonical views may not be easy to infer for non-rigid deformation and when occlusion happens. However, the paper proposes a effective method to alleviate the curse of dimension in 3D when the object is relative simple or symmetric.

Reviewer 2



The paper presents a method for generating high resolution shapes from either a single image or a low resolution shape. Instead of increasing the resolution of the volume, the method is using the depthmaps (ODM) from 6 canonical views (orthographic images from the 6 primary views). The ODMs are estimated from the low resolution shape and then two networks estimate the high resolution depthmaps and the silhouettes of the shape. Finally, the high resolution shape is estimated with 3D model carving. I like the idea of depthmap super-resolution for generating high resolution shapes. Also, the canonical views seem to capture well the details of the shape. Finally, the results look very good and accurate. Some questions: How does the method perform with concave shapes? Does the depth map estimation solve these cases? Regarding the learning of the residual details, was the accuracy of a direct approach inferior? Do you process the depthmaps independently with the same networks? Regarding weaknesses, the use of silhouettes and depthmaps for shape reconstruction is not new, see missing references [A, B, C] (they should be included and discussed). Also the final shape is not estimated directly form the depthmaps but rather from a combination of depthmaps/silhouettes and up-sampled low res shape.Why you can't fuse the depthmaps to get the final shape? [A] Synthesizing 3D Shapes via Modeling Multi-View Depth Maps and Silhouettes with Deep Generative Networks, Soltani et al CVPR 17 [B] Pixels, voxels, and views: A study of shape representations for single view 3D object shape prediction, Shin etal CVPR 18 [C] OctNetFusion: Learning Depth Fusion from Data. Riegler etal 3DV 17

Reviewer 3



This paper proposes a method for super-resolution of a 3D shape represented as voxel grid, e.g. from 32^3 to 512^3. By and large, the method seems well-founded. Retaining just surface information dramatically reduces the complexity of the representation and allows upscaling to large sizes. An obvious issue is incorrect reconstruction of highly-occluded regions (can be mitigated with more views although internal detail can never be recovered). The paper is well-written and the experiments reasonable in justifying the method. I am not aware of other work on (deep learning-based) 3D super-resolution. I am also not familiar with the large existing body of work on 2D super-resolution, so I cannot comment on the novelty of the method compared to related 2D methods, if any.